# PLATO: A Predictive Drug Discovery Web Platform for Efficient Target Fishing and Bioactivity Profiling of Small Molecules

**DOI:** 10.3390/ijms23095245

**Published:** 2022-05-08

**Authors:** Fulvio Ciriaco, Nicola Gambacorta, Daniela Trisciuzzi, Orazio Nicolotti

**Affiliations:** 1Dipartimento di Chimica, Università degli Studi di Bari “Aldo Moro”, Via E. Orabona, 4, I-70125 Bari, Italy; fulvio.ciriaco@uniba.it; 2Dipartimento di Farmacia Scienze del Farmaco, Università degli Studi di Bari “Aldo Moro”, Via E. Orabona, 4, I-70125 Bari, Italy; nicola.gambacorta1@uniba.it (N.G.); daniela.trisciuzzi@uniba.it (D.T.)

**Keywords:** target fishing, bioactivity profiling, multifingerprint similarity searching, drug repurposing

## Abstract

PLATO (Polypharmacology pLATform predictiOn) is an easy-to-use drug discovery web platform, which has been designed with a two-fold objective: to fish putative protein drug targets and to compute bioactivity values of small molecules. Predictions are based on the similarity principle, through a reverse ligand-based screening, based on a collection of 632,119 compounds known to be experimentally active on 6004 protein targets. An efficient backend implementation allows to speed-up the process that returns results for query in less than 20 s. The graphical user interface is intuitive to give practitioners easy input and transparent output, which is available as a standard report in portable document format. PLATO has been validated on thousands of external data, with performances better than those of other parallel approaches. PLATO is available free of charge (http://plato.uniba.it/ accessed on 13 April 2022).

## 1. Introduction

Reducing attrition in the early stage of drug discovery is of utmost importance to prevent expensive flops of improperly designed late-stage products [1] whose fiasco is often due to the inaccurate identification of drug targets with real therapeutic potential. On the other hand, the drug target prediction, also named target fishing, is paramount not only for addressing the design of new active molecules but also for repurposing known drugs and even for optimizing their side effects. Target fishing methods can employ ligand-based and structure-based approaches [2]. The former has returned higher accuracy in prediction as well as quicker results [3]. The basic assumption is that if a pool of molecules is known to bind a given protein drug target, other similar molecules are expected to do the same [4]. Rooted on this idea, we have also conceived and deployed a new statistical method for the quantitative prediction of the bioactivity values of new molecules, that is the bioactivity profiling.

Available for free on the web, PLATO (standing for Polypharmacology pLATform predictiOn) is a technological tool to perform ligand-based protein target fishing and quantitative bioactivity prediction of small molecules. The graphical user interface is intuitive and friendly enough to avoid pitfalls to non-experts. On the other hand, skilled users can interrogate PLATO by getting a wealth of highly structured information.

As in-depth detailed elsewhere [5,6,7], PLATO is based on two predictive algorithms specifically designed for target fishing and bioactivity profiling. Both the algorithms share 13 molecular fingerprints (that are: PubChem, graph, pattern, substructure, cdk_maccs, featmfp1, fp2, rdkit7, klekota_roth, hybridization, mfp1, ap_bits, tt_bits) to compute the 2D Tanimoto similarities on a set of 632,119 compounds having experimentally measured bioactivity data for 6004 protein targets. This amount of data was retrieved from ChEMBL release 30 according to a set of specific filtering rules [8,9]. The interested reader can retrieve more details about molecular fingerprints in our previous published papers [5,6,7].

The present report details how to practically use PLATO. Remarkably, PLATO returns a standard report in portable document format, which includes the list of the top-ranked solutions as well as a wealth of additional information for each single result regarding the ligand chemical structure, the protein drug target and the bioactivity values. The standard report includes hyperlinks to redirect users to ChEMBL for further and deeper investigations. An online tutorial is also available. PLATO is free of charge and available at http://plato.uniba.it (accessed on 13 April 2022).

## 2. Materials and Methods

### 2.1. Technologies

The predictive algorithms behind PLATO are written in Haskell language. Along with a clear and concise algorithm representation, this implementation allows keeping the relevant activity data and several more molecular features in memory, limiting database access and table translation to the initialization phase of the program. PLATO has the capability to treat more than 20 queries per minute and this paves the way to large-scale applications of the method. On demand, virtual reverse screening campaigns of large compound collections can be performed in batch mode through mail requests to the authors.

The fingerprints for all the 632,119 filtered ligands are pre-computed and stored in memory. The fingerprints for the query are computed on request by the frontend. The fingerprints are made available by the RDKit, CDK and OpenBabel libraries through their python bindings [10,11,12].

The web frontend of PLATO was designed to allow both user and programmatic data retrieving through POST requests. The POST form requires only three fields: the SMILES code, the type of computation and the output format. The currently available output formats are pdf report and json encoded data. The web frontend is written in python using the Flask web framework and the Jinja2 templating libraries [13]. A graphical widget, whose aim is to allow users to draw molecules or entering them in various alternative formats (SDF, MOL and InChI key) is featured on the prediction interface; it is made available by the JSME open-source project [14].

The frontend is compatible with any browser and even usable through line commands like curl or wget. As the drawing molecule option might be unusable on very ancient browsers, a text input for the SMILES code is also provided.

### 2.2. Data and Models

Overall, PLATO makes use of a dedicated dataset stored in four tables listing properties and values for targets, ligands, bioactivities and fingerprints.

The dataset is updated frequently and is currently extracted from ChEMBL version 30, based on the application of the following criteria: (i) target filter: “target_type: Single protein | Protein complex”; (ii) ligand filter: “molecule_type: Small_molecule”; “prodrug: not 1”; and (iii) activity record filters: “confidence_score: >5”; “standard_relation: =”; “standard_type: IC50|EC50|Ki|Kd”; “standard_units: nM”; “no comment inherent to inactivity”.

At the time of writing, the dataset consists of 632,119 ligands and 6004 targets, linked by 1,111,534 activity records. All numbers, broken down by species, are given in Table 1. The list of targets is updated and made available in the “DATA” tab of the web site.

### 2.3. Input

The “Prediction” tab contains a user-friendly interface to interrogate the PLATO platform. Users can submit the input query through a graphical widget, either by drawing the 2D structure or by pasting the SMILES notation; MOL and SDF formats are also supported: right-click on the widget to access the relevant menu. The user can then choose the algorithm to employ by flagging “bioactivity profiling” or “target fishing” options. Furthermore, expert users can choose the output format as a json file, by flagging “json data”. Calculations take about 2 to 20 s for each query molecule. All the steps concerning with PLATO “Prediction” are summarized in Figure 1. For completeness, several menu items are also provided in the upper banner including information concerning the quality of data filtered from ChEMBL release 30 (i.e., “DATA” from the menu toolbar). For ease of reference, we report a prediction exercise by employing as a query the chemical structure of a real Cannabinoid Receptor 1 (CBR1) antagonist published only two months ago and thus not yet covered in ChEMBL [15].

### 2.4. Output

PLATO returns as output a standard report in portable document format, which includes the list of the top-scored solutions as well as a wealth of additional information for each single result regarding the ligand chemical structure, the protein drug target and the bioactivity values. The standard report includes hyperlinks to redirect users to ChEMBL for further and deeper investigations. Optionally, predictions can also be retrieved as a json file, containing further details of the computation and suitable for automatic data extraction. Interestingly, a downloadable pdf output is returned based on our multifingerprint similarity algorithms.

#### 2.4.1. Target Fishing Output

As far as the target fishing algorithm is concerned, the query chemical structure and SMILES notation are reported. By default, the output consists of a table containing information about at least 30 predicted targets. However, all the targets predicted as reliable are always shown.

For each predicted target, three main fields are reported as columns:The first column is headed as “Target” and reports the name of the protein target. It can redirect the user to a wealth of additional information;The second column is headed as “score” and reports the value, ranging from 0 to 13, to assess the overall similarity compared to known bioactive ligands for a given protein target;The third column is headed as “reliable” and can return as output “yes” or “not” to indicate if a prediction is accurate or inaccurate.

By clicking on each predicted target, the user is forwarded to a table reporting the most similar bioactive ligands to the query along with the best activity experimental values reported in ChEMBL.

#### 2.4.2. Bioactivity Profiling Output

Likewise for the target fishing, the first page shows the chemical structure and SMILES notation of the query molecule. Moreover, the bioactivity profiling report contains a “summary table” listing five main fields, reported as columns:The first column is headed as ”Target” and reports the name of the protein target. It can redirect the user to a wealth of additional information;Columns 2 to 5 are headed as “IC50”, “EC50”, “Ki”, and “Kd” and report the corresponding predicted bioactivity values;The sixth column is headed as “σ_p_” and reports the best calculated variance for the predicted bioactivity type.

For the bioactivity profiling algorithm, the table reports for each predicted target the most similar known ligands along with the experimental activity values (i.e., IC_50_, K_i_, EC_50_ or K_d_) and is organized as follows:The Target ChEMBL ID can be clicked to automatically open the Target Report Card available in ChEMBL database;The “Structure” column with the ChEMBL identifiers is directed to the corresponding Compound Report Card page, where physiochemical or biological information can be found;The τ value is a precision measure of the predicted bioactivity values.

## 3. Explicative Case Study

An antagonist of the cannabinoid receptor type 1 receptor (CB1R) published just two months ago [15] has been used as a query to practically show how to challenge the predictive potential of PLATO. Noteworthy, CB1R is mostly expressed in the central nervous system and also in peripheral tissues such as pancreas, lungs, liver and ileum, where its inhibition results effective in reducing weight and insulin resistance [16]. The design of this CB1R antagonist was inspired to the structure of the Ibipinabant, a well-known selective CB1R inverse agonist, by replacing the arylsulphonyl and methylamino groups with alkylaminosulphonyl and amidine moieties, respectively, in order to enhance potency and peripheral restriction [17]. Satisfactorily, as shown in Figure 2 and Figure 3, PLATO provided accurate predictions in agreement with the experimental data. Specifically, the target fishing method returned the CB1R as the first reliable target and, on the other hand, the bioactivity profiling predicted an IC_50_ value equal to 65.6 nM, thus providing a good estimate of the experimental measure, which is equal to 1.2 nM. Furthermore, an additional case study concerning with the design of new aromatase inhibitors [18] was provided as Appendix A (see files “Target_Fishing_Case_study_2.pdf” and “Bioactivity_Profiling_Case_study_2.pdf” for outputs of target fishing and bioactivity profiling predictions, respectively). PLATO, again, returned a correct prediction for both methods, identifying the CYP45019A1 as first target, and predicting an IC_50_ value equal to 0.59 µM, very close to the experimental data, equal to 0.86 µM.

## 4. Ligand-Based Reverse Screening

Alongside the graphical interface, PLATO also includes an Application Programming Interface (API) that can be interrogated for the screening large pool of compounds provided as a SMILES list. Specifically, by means of in-house python scripts, the user can choose to carry out the predictions towards all the targets stored in the database, or to constrain the prediction towards a specific target. The obtained outputs are stored in a json format. The ligand-based reverse screening is available on demand.

## 5. Conclusions and Outlook

PLATO provides quick and cheap methods for target fishing and bioactivity profiling and is thus of utmost importance in several applications of drug discovery. At the best of our knowledge, PLATO is the first web platform able to perform bioactivity prediction; noteworthily, the performance of target fishing is parallel or even better than that obtained by other popular webservers [7,19,20]. Noteworthily, PLATO can be effective: (1) for drug repurposing and further optimization studies [21]; (2) to shed light on the mode of action of compounds whose biological counterparts are still unknown; (3) to facilitate the identification of off-targets, thus preventing the occurrence of undesired side effects [22]; and (4) in combination with docking and de novo design studies [23]. At present tailored for small molecules, PLATO could be properly tuned also for dealing with peptide-like compounds and thus opening attractive perspectives for new therapeutic approaches [24].

## Figures and Tables

**Figure 1 ijms-23-05245-f001:**
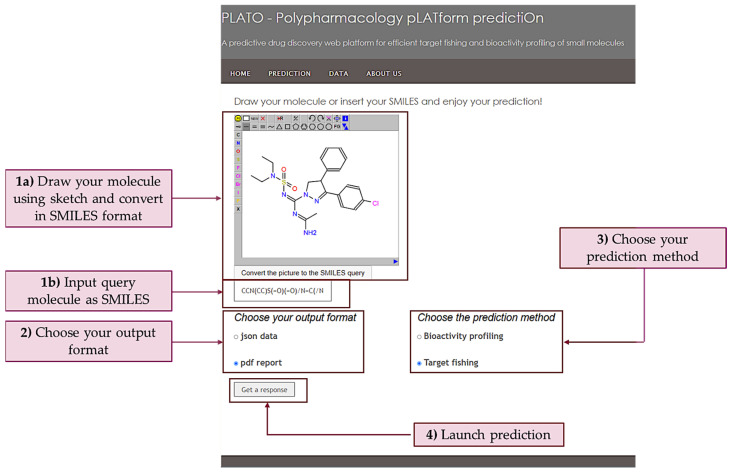
PLATO “Prediction” page. The query molecule is entered either as SMILES or by using the JSME sketcher for opening, importing or modifying a molecular structure. If the query is sketched, the user must click “Convert the picture to the SMILES query” button to translate the 2D structure in SMILES format (**1a,b**), Otherwise, MOL and SDF formats are also supported by right clicking on the widget to access the relevant menu. The user can then choose the output format (**2**) as well as the prediction method, which can be the ‘Target fishing’ or the “Bioactivity profiling” (**3**). The “Get a response” button is used to launch the PLATO prediction (**4**).

**Figure 2 ijms-23-05245-f002:**
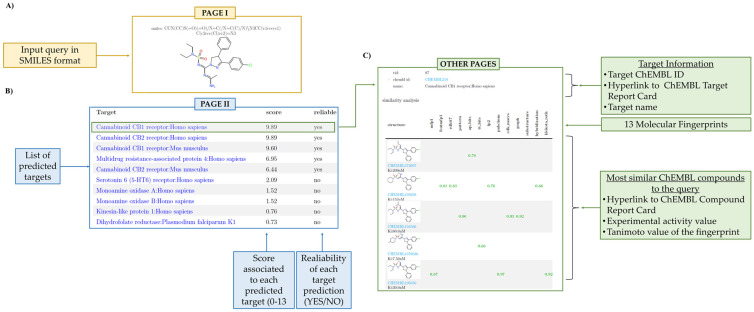
Target fishing output page. (**A**) First page output: the chemical structure and the SMILES of the query. (**B**) Second page output. A table lists the predicted protein targets found by the target fishing algorithm for the given query molecule. By default, protein targets are ranked according to the descending score values (ranging from 0 to 13) to quantify the overall similarity compared to known bioactive ligands for the given protein target. A reliability flag is also returned to indicate if a prediction is accurate or inaccurate. (**C**) Other pages. Each predicted protein target is redirected to the “similarity analysis” page. For each target, the Target ID and the associated ChEMBL Target Report Card are provided; a table reports the most similar known ligands along with the best experimental bioactivity value and the Tanimoto similarity values based on 13 molecular fingerprints. A green/red flag indicates over/under-threshold similarity values. Hyperlinks to the ChEMBL Compound Report Card for each similar known ligand are also provided.

**Figure 3 ijms-23-05245-f003:**
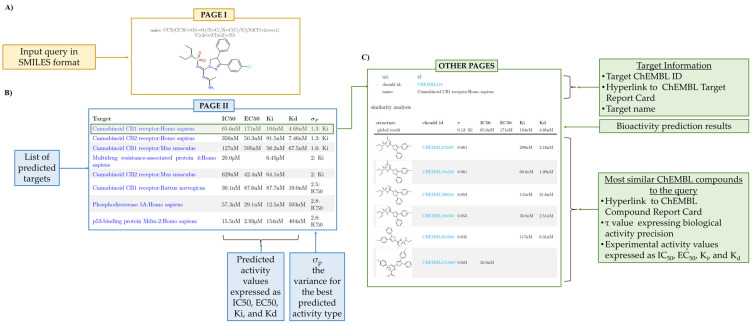
Bioactivity profiling output page. (**A**) First page. The chemical structure and the SMILES of the query. (**B**) The second page. A table lists the predicted protein targets found by bioactivity profiling algorithm for the given query molecule. The predicted bioactivity values are expressed as IC_50_, EC_50,_ K_i_ and K_d_. By default, target proteins are ranked according to the σ_p_ value, which is the best variance for the reported predicted bioactivity types. (**C**) Other pages. Each predicted protein target is redirected to the ”similarity analysis” page. For each target, the Target ID and the associated ChEMBL Target Report Card are provided; a table reports the most similar known ligands with the best experimental bioactivity values along with the τ value that is a measure of precision. Hyperlinks to the ChEMBL Compound Report Card for each similar known ligand are also provided.

**Table 1 ijms-23-05245-t001:** Volumes of data extracted from ChEMBL version 30 for target fishing and bioactivity profiling.

	Homo Sapiens	Rattus Norvegicus	Mus Musculus	Other
Number of targets	2840	736	645	1783
Number of interactions	866,306	95,545	30,030	119,653

## Data Availability

Not applicable.

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
