# Peer review of "PLATO: A Predictive Drug Discovery Web Platform for Efficient Target Fishing and Bioactivity Profiling of Small Molecules"

_ijms, 2022, doi:10.3390/ijms23095245_

Round 1
Reviewer 1 Report
The purpose of the report is to explain how to use the PLATO software. The paper is well written and easy to read.
Author Response
The purpose of the report is to explain how to use the PLATO software. The paper is well written and easy to read.
We thank the reviewer for her/his comment.
Reviewer 2 Report
The body of work reported here bY Ciriaco et. al. may be a fantastic computational tool but is not appropriate for the IJMS. This work purely relies on the development and validation of algorithms and refinement of such models. The databases of pharmacologically active compounds and targets are not well defined or extremely large.
I strongly suggest that this study is not suitable for this journal and that this work be submitted to a Computational pharmacology journal (ACS J of Chem Infm and Modelling ....etc) for review and consideration
Author Response
The body of work reported here bY Ciriaco et. al. may be a fantastic computational tool but is not appropriate for the IJMS. This work purely relies on the development and validation of algorithms and refinement of such models. The databases of pharmacologically active compounds and targets are not well defined or extremely large.
I strongly suggest that this study is not suitable for this journal and that this work be submitted to a Computational pharmacology journal (ACS J of Chem Infm and Modelling etc) for review and consideration.
We thank the reviewer for her/his very nice comments. Please consider that both algorithms for target fishing and bioactivity prediction have been recently published in J. Chem. Inf. Model. 2021, 61, 4868–4876, doi:10.1021/acs.jcim.1c00498. and J. Chem. Inf. Model. 2019, 59, 586–596, doi:10.1021/acs.jcim.8b00698]. In this paper, we have preferred to promote both the algorithm crafted an integrated tool and available as a free and easy web platform. This will indeed inspire users to use our drug discovery tools for practical purposes.
Reviewer 3 Report
Comments to the author(s):
This manuscript reports user instructions for a web-based platform, PLATO (Polypharmacology pLATform predictiOn), developed for target fishing and bioactivity profiling for drug discovery applications based on two predictive algorithms that are reported elsewhere by the group. The organization and writing for this manuscript is clear, but my comments/suggestions should be addressed.
- The authors state PLATO ‘Prediction’ page supports MOL and SDF for query molecules. However, Figure 1 does not show how the MOL or SDF file of a molecule can be imported for prediction.
- Details for 13 molecular fingerprint types used for similarity searches are missing. Also, the authors should provide a list/name of targets that are available for screening in PLATO.
- Images in Figure 1 and Figure 2, specifically corresponding to the output pages are not clearly visible. As these referred to the case study used to show the prediction strength of PLATO. I would suggest the author should use high-resolution images to make it more clear to readers.
- The authors should consider including one more case study to evaluate the prediction efficiency provided by PLATO.
- The authors should mention how PLATO is efficient/different than the other well-known similar drug discovery tools.

Author Response
Comments to the author(s): This manuscript reports user instructions for a web-based platform, PLATO (Polypharmacology pLATform predictiOn), developed for target fishing and bioactivity profiling for drug discovery applications based on two predictive algorithms that are reported elsewhere by the group. The organization and writing for this manuscript is clear, but my comments/suggestions should be addressed.
- The authors state PLATO ‘Prediction’ page supports MOL and SDF for query molecules. However, Figure 1 does not show how the MOL or SDF file of a molecule can be imported for prediction.
We thank the reviewer for her/his suggestion. In this respect, we added a sentence in the main text to better clarify this point.
Details for 13 molecular fingerprint types used for similarity searches are missing. Also, the authors should provide a list/name of targets that are available for screening in PLATO.
We thank the reviewer for her/his comment. In this regard, the list of the 13 molecular fingerprints has been included in the main text of the revised manuscript whereas the list of targets is available in the ‘DATA’ tab of the website (http://plato.uniba.it/). Additional information about the 13 molecular fingerprints is also detailed in our previous works [J. Chem. Inf. Model. 2021, 61, 4868–4876, doi:10.1021/acs.jcim.1c00498.; J Chem Inf Model 2019, 59, 586–596, doi:10.1021/acs.jcim.8b00698].
Images in Figure 1 and Figure 2, specifically corresponding to the output pages are not clearly visible. As these referred to the case study used to show the prediction strength of PLATO. I would suggest the author should use high-resolution images to make it more clear to readers.
We thank the Reviewer for this comment. Following her/his suggestions, Figure 1 and Figure 2 have been modified accordingly.
The authors should consider including one more case study to evaluate the prediction efficiency provided by PLATO.
We thank the reviewer for this observation, which has contributed to improve the quality of the revised manuscript. As suggested by the reviewer, another case study has thus been added as Supporting Information. Hopefully, PLATO, again, returned efficient predictions based on both algorithms. Please see files ‘Target_Fishing_Case_study_2.pdf’ and ‘Bioactivity_Profiling_Case_study_2.pdf’ for outputs of target fishing and bioactivity profiling predictions available as Supporting Information.
The authors should mention how PLATO is efficient/different than the other well-known similar drug discovery tools.
We thank for this comment. However, please consider that this analysis was already performed in our previous paper successfully comparing our algorithms with those of two other well-known drug target prediction web platforms that are Polypharmacology Browser and SwissTargetPrediction